# Emotional Reasoning and Psychopathology

**DOI:** 10.3390/brainsci11040471

**Published:** 2021-04-08

**Authors:** Amelia Gangemi, Margherita Dahò, Francesco Mancini

**Affiliations:** 1Department of Cognitive Sciences (COSPECS), University of Messina, Via Concezione, 6/8, 98121 Messina, Italy; margherita.daho@unime.it; 2Department of Psychology, Telematic University of Rome “Guglielmo Marconi”, Via Plinio 44, 00193 Rome, Italy; mancini@apc.it; 3Associazione di Psicologia Cognitiva, Viale Castro Pretorio, 116, 00185 Rome, Italy

**Keywords:** emotion, beliefs, emotional reasoning, affect-as-information, psychopathology, anxiety, guilt, obsessive-compulsive disorder, anxiety disorders, depression, personality disorders

## Abstract

One of the several ways in which affect may influence cognition is when people use affect as a source of information about external events. Emotional reasoning, ex-consequentia reasoning, and affect-as-information are terms referring to the mechanism that can lead people to take their emotions as information about the external world, even when the emotion is not generated by the situation to be evaluated. Pre-existing emotions may thus bias evaluative judgments of unrelated events or topics. From this perspective, the more people experience a particular kind of affect, the more they may rely on it as a source of valid information. Indeed, in several studies, it was found that adult patients suffering from psychological disorders tend to use negative affect to estimate the negative event as more severe and more likely and to negatively evaluate preventive performance. The findings on this topic have contributed to the debate that theorizes the use of emotional reasoning as responsible for the maintenance of dysfunctional beliefs and the pathological disorders based on these beliefs. The purpose of this paper is to explore this topic by reviewing and discussing the main studies in this area, leading to a deeper understanding of this phenomenon.

## 1. Introduction

What maintains dysfunctional beliefs and psychological disorders as well as their resistance to change is a topical question for clinical and cognitive scientists. The explanation of these phenomena is especially fundamental to any theory of pathological suffering and thus is also fundamental to clinical cognitivism. In this paper, we will try to answer these crucial questions, examining the influence of emotions on some cognitive processes and how this can strengthen the beliefs at the basis of the emotional experience itself. In this perspective, we will refer to the appraisal theories of emotions. Specifically, the appraisal-based theories attest that both normal and abnormal emotional states are based on “a person’s subjective evaluation or appraisal of the personal significance of a situation, object, or event on a number of dimensions or criteria” [1] (p. 637). What we want to address here is the influence of emotions on beliefs (i.e., representations, cognitions, assumptions) and how this could lead to the maintenance of pathological disorders.

## 2. Emotional Reasoning or Affect-as-Information

From cognitive psychology, we know that affective-emotional dispositions influence cognitive processes in different ways; all, however, reinforce the assumptions that support that specific emotional state. According to Caprara and Cervone [2], for example, in all people, the affective state increases the availability of images, thoughts, and memories congruent with the previous emotional state (priming effect), even if the emotional state depends on events belonging to different situations (e.g., listening to sad and melancholy music tends to induce sad and gloomy thoughts or images in our minds, even if we were fine until a moment before). Yet another of the numerous ways in which affect can influence cognition is when people use affect as a source of information about external events [3,4]. Emotional reasoning, ex-consequentia reasoning, and affect-as-information are indeed terms referring to a particular psychological mechanism that can potentially lead individuals to take their emotions as information of external events, even when the emotion is not generated by the situation to be evaluated. The theory of emotional reasoning or affect as information describes, therefore, the psychological mechanism according to which human beings tend to use their emotions as significant information to express evaluations and judgments about the world rather than refer objectively to reality.

This mechanism differs from the "priming effect", since in this case, the influence of emotions on cognitive processes is indirect; in the case of affect-as-information, the influence of emotions on the evaluations expressed by the individuals is more direct. In fact, emotional states have the power to influence people’s judgments, especially when they are experienced as providing judgment-relevant information [5,6,7]. When people make evaluative judgments, they may question how they feel about an event, an activity, or a topic to be evaluated [6]. However, when individuals experience these circumstances, they find it hard to distinguish the affective responses to the event (how they feel about it) from the pre-existing affect that can be considered irrelevant to the event (how they feel at that particular moment). Thus, the hypothesis is that pre-existing emotions may bias evaluative judgments of unrelated events or topics [8]. For instance, Schwarz and Clore [4] showed that people who were asked to describe a negative episode in their lives reported a lower level of life satisfaction compared to those who wrote about a positive life event.

Gasper and Clore [8] and Scott and Cervone [9] deeply explored the affect-as-information mechanism in healthy subjects, finding that participants used laboratory-induced negative affect as information for performance standards (the level of performance adopted by people as a personal standard for evaluating their achievement) and for judgments of risk (likelihood and severity of a negative outcome). In particular, Scott and Cervone [9] showed that negative affect can influence the construction of higher performance standards even if the nature of the performance is unrelated to the source of the negative affect. Moreover, Gasper and Clore [8] showed that negative affect influences the evaluation of risks. Indeed, participants that underwent an induced negative affect (such as sadness) estimated both personal and impersonal negative events as more likely and severe than the participants that received an induced positive affect. Most of the research with a non-clinical sample has thus highlighted how affective-emotional dispositions actively influence some basic cognitive processes, albeit indirectly. In return, this mechanism seems to feed vicious circles responsible for strengthening the beliefs underlying the affective state itself.

Accordingly, emotion can work as a source of information, especially in people who experience that emotional state chronically. From clinical observations, some authors have maintained that the tendency to infer danger based on subjective anxiety (ex-consequentia reasoning) may have a role in the development and maintenance of anxiety disorders by starting a vicious circle that can be summarized in this way: anxiety induces a sense of threat, which further stimulates more anxiety, and so on. The first and pioneering studies in the clinical field were those by Arntz [3], which demonstrated that the affect-as-information theory may function as a mechanism for maintaining anxiety and mood disorders. Nowadays, the literature suggests that this phenomenon is more likely to be noticed in clinical populations because of its endemic negative affect [10]. Indeed, the exaggerated use of negative feelings as a source for evaluating danger (i.e., emotional reasoning) has been revealed to have an important clinical relevance both as a maintenance factor of a disorder and as a treatment target in cognitive therapy [11].

However, despite the substantial evidence and numerous studies about how affect influences cognition in normal subjects [12,13], too little is known about the population suffering from psychological disorders. The importance of emotions (e.g., anxiety) and cognition in the development and maintenance of psychopathological disorders has been emphasized for both adults [11] and children [14]; however, there are still questions regarding the relationship between maintaining dysfunctional beliefs or a pathological condition and the use of emotion as information. In light of these considerations, the aim of this paper is to explore this topic by reviewing the most relevant findings in psychopathology.

### Feelings-As-Embodied Information

More recent studies suggest that affect-as-information can also be “embodied”. The embodied cognition hypothesis on emotions introduces three new implications, which are important for understanding the interaction between affect-as-information or emotional reasoning and maintenance of clinical disorders. The first implication is that off-line emotions may also take the form of “mental images” [15]. In particular, the vividness of the image, which depends on the accessibility of affective systems, may influence the interaction of emotions with judgments, contributing to the development and maintenance of emotional disorders. The second implication is that emotional states do not disappear after they are experienced, given that they can be re-created without external stimuli. Indeed, the activation of the image of an emotional situation plays the role of “as if real” templates, which trigger emotional reactions and feelings [16]. The last implication is that the embodied simulation of emotions seems to re-use the same cognitive processes involved during a simulated emotional experience [17,18]. Because of this mechanism, the differences in the affective systems re-used for simulation (such as hyper-reactivity in anxiety) may be reflected in the differences of the simulations (e.g., the ability to elicit emotion, the vividness of the images, etc.) and their influence on the emotion.

Therefore, according to the embodied approach, emotional states can be co-opted as embodied simulations of emotions or as a partial re-creation of an emotional state. The outcome is that anxious individuals may perceive more danger because they assess it based on highly vivid mental images of threatening or catastrophic events and anxious feelings [18,19,20,21,22]. This mechanism has been called “simulation heuristic” [23,24]. Anxious individuals are also characterized by hyper-reactivity of affective systems [25,26], and thus, they may simulate more vividly feelings of anxiety to assess danger, judge situations, or cope with events. Unfortunately, the risks of this cognitive bias include both hampering the identification of false alarms and to behaving in a way to confirm the a priori threat value of the feared stimuli/situations.

## 3. Emotional Reasoning or Affect-As-Information Mechanism in Anxiety Disorders

Several studies show that anxious individuals tend to easily engage in emotional reasoning, leading them to draw invalid conclusions about a situation based on their subjective emotional assessment or response [3,27]. The mechanism of emotional reasoning has been clearly described by Beck, Emery, and Greenberg [28]: “many anxious patients use their feelings to validate their thoughts and thus start a vicious circle: I’ll be anxious when I ask for the date, so there must be something to fear’’ (p. 198). Arntz et al.’s [3] experiments on emotional reasoning show that adult anxious patients inferred danger based on their anxious response, whereas normal controls inferred danger primarily based on objective information. In particular, in their first experiment, Arntz et al. [3] compared three groups of patients with anxiety disorders (spider phobia, panic disorder, and social phobia) to a group of individuals with a mixture of anxiety disorders (i.e., obsessive-compulsive disorder (OCD), generalized anxiety disorder (GAD), or post-traumatic stress disorder (PTSD)) and a non-clinical control group to evaluate responses to an emotional reasoning task. Participants were asked to rate the danger evident for each of a series of different scripts according to objective danger (i.e., whether or not objective danger or objective safety information was included) and emotional response (i.e., whether or not an anxious or non-anxious response was indicated in the script). The scripts also varied according to disorder relevance. The panic-disorder-relevant script described being in a crowded elevator. Participants were provided with four possible endings to this scenario, as follows (p. 919):“One of the passengers suddenly falls into your arms. You smile. You have been interested in this person for quite some time and this seems to be a good opportunity" (i.e., objective safety and non-anxious response).“Suddenly you become very anxious” (i.e., objective safety and anxious response).“All of a sudden the elevator gets stuck between floors. The ventilator stops and the elevator won’t budge. You see two people faint: one falls into your arms. You smile. You’ve been interested in this person for quite some time and this seems to be a good opportunity” (i.e., objective danger and non-anxious response).“All of a sudden the elevator gets stuck between floors. You have seen two people faint. Suddenly you become very anxious” (i.e., objective danger and anxious response).

Participants were considered to have engaged in emotional reasoning if their danger ratings for a given script were greater when the ending included an anxious response compared with a non-anxious response (averaged across objectively safe and dangerous endings). Arntz et al. [3] reported that anxious patients engaged in emotional reasoning, as evidenced by significantly greater danger ratings for scripts that contained anxiety response information than for those without this information. In contrast, the non-clinical participants provided only marginally more negative ratings for scripts that described an anxious response. Interestingly, there were no significant differences across anxiety disorder diagnostic groups in the extent to which they exhibited emotional reasoning. Moreover, the emotional reasoning tendency was general rather than disorder-specific; anxious patients demonstrated emotional reasoning across all scripts, not just those that were thematically linked to their disorder. The authors maintained that this tendency to infer danger based on subjective anxiety (ex-consequentia reasoning) may play a role in the development and maintenance of anxiety disorders by starting a vicious circle in which anxiety leads one to infer that there is a threat, which further stimulates anxiety, which further increases the sense of threat, and so on. There is also evidence to suggest that emotional reasoning may play a role in PTSD [29]. Engelhard et al. [29] administered three of Arntz et al.’s [3] scripts (relating to panic disorder, social phobia, and spider phobia) to Vietnam combat veterans with and without PTSD. Overall, the veterans with PTSD provided greater danger ratings than those without PTSD when the subject of these three scripts was described as having an anxious response compared with a neutral response.

Emotional reasoning has also been investigated in children using similar experimental procedures to those employed by Arntz et al. [3]. These studies have provided evidence that, while almost all children engage in emotional reasoning to some extent [30,31,32,33,34,35,36], it may be the persistence of this tendency into adulthood that confers an increased risk of subsequent psychopathology [30,31]. This contention is based partly on the finding that emotional reasoning, although common in all children, is negatively correlated with cognitive development in 10- to 13-year-olds and partly on the finding that healthy adults in Arntz et al.’s [3] study did not appear to engage in emotional reasoning. 

Consistent with the findings of studies that have used adult samples [3], Morren et al. [30] found that primary school children demonstrated emotional reasoning tendencies across three different anxiety-themed vignettes (about social, generalized, and separation anxiety). Together with the findings from adult samples [3], these results suggest that emotional reasoning may be a general tendency that is not specific to particular situations. However, central to these models is the hypothesis that interpretation biases, such as the over-estimation of danger in the environment, provoke anxious feelings and drive avoidant behaviors, which then reinforce those same interpretations.

## 4. Emotional Reasoning or Affect-As Information Mechanism in Obsessive-Compulsive Disorder (OCD)

Emotional reasoning has been invoked also to explain the maintenance of beliefs characterizing OCD. A first hypothesis, named mood-as-information, has been cited by several authors, including Clark [37], MacDonald and Davey [38], O’Connor [39], and Rachman [40,41], to explain OCD. However, MacDonald and Davey [38] were the first to empirically explore this hypothesis. The authors maintained that the perseveration of a disorder or symptom occurs whenever people experience a negative mood and ask themselves “Did I do as much as I can with this task?”. In their study, they also showed that healthy subjects checked more while correcting a text with numerous errors when they questioned themselves using the “Did I do as much as I can?” stop rule. Similarly, healthy participants also reported more checks when they were in a negative mood compared to participants who were in a negative mood but used another stop rule (“Do I feel like continuing?”). While MacDonald and Davey [38] interpreted these findings as support for the “mood as input” explanation in compulsive checking, other results call into question this conclusion [13]. Indeed, other authors objected to the fact that the experimental “perseveration” observed in the MacDonald and Davey [38] paradigm occurred only for complex tasks (such as checking a complicated text) and that the “perseveration” was functional for participants’ reasoning (such as to detect errors), while compulsive perseveration occurred typically for very simple tasks, and it was inherently non-functional. In the last case, task performance did not increase with repetition. In fact, MacDonald and Davey’s [38] results disappeared once the task request was easier and more OCD-relevant. Thus, it may be doubted whether the application of the “mood-as-input” mechanism works as a valid model of OCD checking.

Later, another explanation was proposed for how emotional reasoning might be relevant to understanding OCD [27]. A significant body of literature showed that the fear of guilt for having acted irresponsibly, with an exaggerated sense of responsibility, appears to be central to the appearance, development, and perpetuation of obsessions and compulsions [40,41,42,43,44,45,46]. Therefore, according to this hypothesis, obsessions and compulsions could be considered as activities aimed at preventing the anticipated experience of guilt for having acted irresponsibly. That is, OC perseverations may follow from the fear of behaving irresponsibly, causing unjustified damage to oneself or others, and/or violating a moral norm [42,47,48,49,50]. Some authors thus questioned whether the experience of guilt was also related to the increased sense of threat and the decreased belief in the effectiveness of normal washing, checking, or other preventive activities. Gangemi, Mancini, and van den Hout [27] were the first to test this hypothesis. In their study, in line with earlier investigations on emotional reasoning, they explored whether the feeling of guilt might, in and of itself, increase the sense of threat, reducing the satisfaction of preventive action even when the source of the guilt experienced has nothing to do with the situation for which the threat is to be evaluated. Accordingly, compulsive perseveration would result from the obsessive patient’s goal of avoiding the risk of not complying with one’s own perceived responsibilities. In their study, participants were divided into high- and low-trait guilt groups based on a trait measure of guilt. In both groups, the feeling of guilt was induced by asking participants write about a guilt-related, anxiety-related, or neutral life event. Later, participants were assigned to one of three affect induction conditions (guilt, anxiety, or neutral) to ensure that the possible effects of induced guilt were due to state guilt and not to negative affect in general. The emotional states were thus neither generated by nor related to the task used later in the experiment. The task involved requesting participants to estimate both the likelihood and severity of a negative outcome and the dissatisfaction with preventive performance for scripts related thematically to OCD. Thus, individuals were asked to consider two situations, which, if left unchecked, may cause harm or negative outcomes for which they may evaluate themselves as guilty for having acted irresponsibly [42]. Participants were presented with scenarios such as the following: “You are at your home together with your friends. Your parents are away. You and your friends decide to go to a pub to join other friends. You and your friends leave your house while you are playing around and joking. Later, while staying in the pub, it strikes you that you might not have checked the shutters and that the house might have been burgled”.

Furthermore, after having read the scenarios, participants completed a questionnaire to assess the likelihood of the negative event occurring (for example, “How likely is it that burglars had broken into your home?”) and to estimate the severity of the negative event (for example: “How severe would the harm suffered by your family be in the case of housebreaking?”). Results demonstrated that guilt had specific effects on threat estimates. However, although guilt and anxiety share a similar negative valence, only guilt induction led participants to estimate the negative event as more severe and more likely than “anxious” participants. Moreover, the authors found that guilt also had specific effects on performance standards. Guilt induction led participants to estimate their performance aimed to prevent a negative outcome as more unsatisfactory compared to participants of the anxiety-induction group. Therefore, these findings suggest that OCD patients may use the feeling of guilt as information; however, the debate is still open, as this question awaits testing with clinical participants. 

To demonstrate that anxiety, together with responsibility, could be used as information, leading to an overestimation of threat in obsessive patients, [51] Lommen and colleagues conducted an exploratory study. They examined emotional reasoning based on feelings of anxiety in a group of patients affected by OCD and compared this to a group of patients with other anxiety disorders as well as a healthy control group. They also assessed if these groups were influenced by feelings of responsibility in judging the dangerousness of situations. Their findings showed that OCD patients used emotional reasoning comparable to the anxiety patients’ group. However, in contrast to previous results, they also found that healthy controls use emotional reasoning, though this seemed to be restricted to situations that contained objectively safe information. However, in line with Gangemi et al.’s findings [27], only the OCD group rated the situation as more dangerous when feelings of responsibility were present; the anxiety patients and healthy controls did not seem to be influenced by feelings of responsibility. In summary, it can be hypothesized that feelings of guilt and feelings of anxiety, though only those associated with responsibility, are associated with an increase in the perceived likelihood and severity of OCD-relevant risks. Thus, individuals with OCD might also engage in emotional reasoning, given that emotional states such as anxiety, responsibility, and guilt seem to contribute to increased levels of perceived risk and high standards.

However, questions remain regarding emotional reasoning and the obsessions and compulsions that are not related to fear of guilt or responsibility. Some studies have shown that disgust-based reasoning can contribute to the development and perpetuation of obsessions and compulsions, like cleaning rituals, that are linked to the fear of contamination [52]. According to the idea that contamination-fearful individuals are typically characterized by a heightened propensity for disgust as a trait [52,53,54], it has been hypothesized that the enhanced tendency toward fear of contamination would lead individuals to experience disgust in any given situation, increasing the presence of disgust-based emotional reasoning and its threat-confirming influence. In a recent study by Verwoerd et al. [55], the authors tested empirically whether disgust-based reasoning might be involved in fear of contamination. Based on the contamination fear subscale of the Padua Inventory (PI), the authors selected two groups of non-clinical participants, scoring lower and higher than the established clinical range. Both groups were presented with a series of scripts describing everyday contamination-relevant scenarios (e.g., walking on the road and detecting a urine smell; touching a parking meter), which systematically varied in terms of the absence/presence of the actor’s disgust response and the absence/presence of an objective threat of contamination. They were then asked to report their perceived threat of contamination or illness. As expected, high contamination-fearful individuals inferred the occurrence of a threatening outcome based on their disgust response, in addition to being based on an objective threat. Emotional reasoning was thus conceptualized as a more general overestimation of danger, a higher subjective probability of contracting a disease, and an enhanced risk of becoming contaminated.

## 5. Emotional Reasoning or Affect-As-Information in Mood Disorders

According to the literature, depressive patients may use their negative feelings to validate their thoughts, starting vicious circles that may contribute to their depression being maintained [9]. However, the relationship between emotional reasoning or affect-as-information and depression has not been systematically studied using clinical samples. Scott and Cervone [9], after having demonstrated that negative affect can lead to the construction of higher performance standards in non-clinical groups, argued that their results could have relevant implications for the understanding and treatment of depression. They suggested that depressed people may use their negative feelings to validate their thoughts, starting a vicious circle that contributes to their depression being maintained. The depressed mood could thus increase performance standards, thus decreasing the chance that the standards are met, which decreases mood, which increases standards, and so on. However, their hypotheses, although very reasonable for those who have experience with depressed patients, have never been empirically tested. 

Berle and Moulds [56] were the first to test empirically the hypothesis that depression is characterized and maintained by elevated levels of emotional reasoning. Their experiment was aimed at determining whether people who were experiencing a current major depressive episode engaged in emotional reasoning to a greater extent than those who were not depressed. Furthermore, they tried to verify whether previously depressed individuals engaged in higher levels of emotional reasoning when compared with a group of never-depressed individuals, to provide insights into whether elevated levels of emotional reasoning are confined to depressive episodes or whether they might persist following an episode. According to the authors, if important levels of emotional reasoning persist between different episodes, it may indicate that emotional reasoning can serve as a risk factor or marker for individuals who later develop depression, that emotional reasoning is a more enduring trait-like process [3], or that the tendency persists as some sort of “cognitive scar” [57].

To this aim, individuals who were currently experiencing a major depressive episode were compared with a group of non-depressed and never-depressed individuals in performing an emotional reasoning task. The task was an adapted from Arntz et al. [3] and included seven scenarios. Participants were asked to vividly imagine themselves in the situation described in the scenarios presented, each of which had four different endings: (1) objectively neutral and with a non-valenced emotional response, (2) objectively neutral with a dysphoric (or anxious) emotional response, (3) objectively negative and with a non-valenced emotional response, and (4) objectively negative and with a dysphoric (or anxious) emotional response. For this task, they also used the ratings from Arntz et al.’s study [4]: how dangerous is the situation, how safe is the situation, how controllable is the situation, how unpleasant is the situation, and how good is this outcome.

The study of Berle and Moulds [56] showed that emotional reasoning tendencies may not be conspicuous in currently depressed individuals. When the depressed and non-depressed participants in their experiment were compared, the first group scored higher on each of the emotional reasoning ratings than the group of non-depressed individuals; however, these differences were not found to be statistically significant. Similar data was found in the comparison between previously and never-depressed participants: even though the mean scores for the previously depressed participants were consistently higher than those for the never-depressed group, the difference was not significant. According to the authors, this might be because depressed individuals vary greatly in the degree to which they engage in emotional reasoning, which may be related to the level of severity of their symptoms. Moreover, individuals with remitted depression may show higher levels of non-self-referent emotional reasoning than those who never suffered from a depressive episode; that is, they rely on their emotions when forming interpretations about situations. Finally, according to Berle and Moulds [56], another possibility is that anxious individuals have a greater awareness of emotional state information, because of their hypersensitivity to their internal and external environments, than depressed individuals. Thus, anxious individuals may be more likely to allow emotions to pervade their interpretations of situations, thereby allowing engagement in emotional reasoning, than depressed patients. Previous studies indeed found that elevated levels of emotional reasoning were associated with an anxiety disorder [3,29]; however, the results of this study suggest that this may not be the case for depression.

## 6. Emotional Reasoning and Personality Disorders

Personality disorders are often considered among the most difficult to be treated as they often require longer therapies and a greater expenditure of therapist energy. Despite therapeutic effort, patients may show little progress because of the substrate of powerful beliefs that constantly generate difficulties in developing coping skills. Another specific characteristic of personality disorders is that they both result in and are a result of the individual creating an egosyntonic template for viewing the world. Due to the inability to look at their responses critically, patients seem also to be subjected to emotional reasoning [58]. While classic clinical cognitivist models underline how the information-processing bias in patients with personality disorders distorts the individual’s perception, attention, thoughts, beliefs, expectations, and memory in virtually all important areas of their lives, little attention has been paid to the central role of emotional reasoning in maintaining these kinds of disorders.

One of the key features of the application of the emotional reasoning hypothesis to perseverative psychopathologies is that most of the disorders exhibit negative mood (such as anxiety, sadness, or fear), which provides information about whether the goals of the individual’s symptomatic behavior have been met. For instance, a recent study by Masland and colleagues [59] examined the influence of emotional reasoning on trustworthiness appraisals in individuals with borderline features. Participants were divided into two groups: individuals with three or more borderline personality disorder symptoms and individuals experiencing two or fewer symptoms. Participants were first exposed to negative, neutral, and positive emotional information. They were then asked to rate the trustworthiness of unfamiliar faces. More specifically, after have completed an affective priming task, in which participants viewed a series of negative/threatening, neutral, or positive images, a photograph of an unfamiliar face (male or female) was presented, and participants were asked to rate the trustworthiness of the faces on a seven-point scale where plus three points corresponded to untrustworthy and minus three to trustworthy, while zero is neutral (scenes and faces were not related and facial expressions were natural and neutral). Results show that participants with suspected borderline pathology made more untrusting appraisals and they were more influenced by negative primes compared to the control group. No differences between the two groups were found for the influence of positive or neutral affective primes. Thus, individuals with borderline features may be differentially and specifically influenced only by negative affective information. According to the authors, a negative emotional context may thus augment biases in the interpretation of social cues (including in the therapeutic setting) and may be related to mechanisms that support, for example, the transient stress paranoia experienced by these patients. Moreover, according to the Emotional Cascade Model of borderline personality disorder [60], emotional events usually trigger rumination and negative cognition, which then increases the emotional intensity and drives behavioral dysregulation. This behavioral dysregulation results in negative consequences, including interpersonal conflicts, which in turn only fuels further “emotional cascading” and rumination.

Emotional reasoning might thus impact patients with personality disorders and have a critical role in maintaining the disorder as well as behavioral dysfunctions. Indeed, another possible explanation of the mechanism is that individuals affected by personality disorders show greater amygdala reactivity to stimuli of various types, and thus they have more problems with the cognitive control of negative emotional information. In particular, prefrontal areas exert inefficient regulatory control of the amygdala in those affected by borderline personality disorder [61,62,63]

## 7. Conclusions

### 7.1. Why Does Emotional Reasoning Affect Only in Some of Us?

At the end of this review on emotional reasoning, it seems useful to raise a relevant question for a deeper understanding of individual differences and the maintenance of psychological disorders: the main object of this paper. Why does is affective state used as a reliable source of information only in some individuals, often feeding vicious circles that reinforce the beliefs at the base of the affective state itself? The answer comes from some of the studies mentioned above [8,27,52], which have explicitly investigated the role of emotion experienced chronically (for example, trait anxiety, trait guilt, trait fear/phobia of contamination, or disgust propensity) in the use of state-emotion, instead of emotions experienced temporarily, for judgments and evaluations. According to the findings of these studies, state-emotion assumes greater relevance if supported by previous information provided by the same trait-emotion.

On the other hand, the state-emotion may not assume any relevance if the information it produces is inconsistent with that generated by the trait-emotion. Therefore, trait-emotion can be considered responsible for individual differences in the use of state-emotion as a source of information. Individuals with high trait-emotion tend to have greater confidence in the information provided by their temporary emotional state. For example, Gasper and Clore [8], after having shown that negative affect influences risk estimates in healthy participants (see above), also found a relationship between trait-affect and emotional reasoning, demonstrating the role of trait-affect in people’s use of state-affect when making judgments of risk. Indeed, after a negative induction, high-trait anxious people estimated more risk, while low-trait anxious participants did not. Moreover, Gangemi et al. [27] found that trait guilt influences the way state guilt is used as information on the judgment of risk (likelihood and severity of a negative outcome) and on the evaluation of preventive performance. In particular, they found that the guilt induction had a relevant role, as it led individuals to evaluate a negative event as more severe and more possible than deemed by ‘‘anxious’’ participants and that these findings were qualified by an interaction with trait guilt. Furthermore, high-trait guilty individuals estimated the likelihood and severity of the negative event only after the guilt induction, not after the anxiety and neutral affect induction, while low-trait guilty individuals did not show this pattern. Finally, they noticed that high-trait guilty individuals reported higher levels of dissatisfaction with their preventive actions after the guilt induction than after both an anxiety and a neutral affect induction, whereas low-trait guilty participants did not. Thus, it seems that high-trait guilt influences the way temporary affect is used as information on the judgment of the severity and the likelihood of a negative outcome and on the estimation of preventive performance, while in the low-trait guilt group the state affect had no effect. All these findings thus suggest the existence of a relevant difference between individuals scoring high and low in trait anxiety or guilt; this may be due to the faith in the informational value of anxious or guilty feelings.

In addition, these experiments clearly show that (a) state -and trait-emotion represent two distinct sources of information; (b) state- and trait-emotion are related; (c) emotion experienced chronically appears to be more reliable in guiding evaluations and judgments. The latter point recalls what Damasio [64] stated in his book, Descartes’ Error (1994): the emotion experienced chronically (dispositional affect) amplifies the informational effect of the experienced emotion transiently.

### 7.2. How Does Emotional Reasoning Lead to the Maintenance of Psychological Disorders?

We want to conclude with a brief review on why affect as an information mechanism leads to the maintenance of some pathological disorders, such as anxiety disorders, mood disorders, obsessive-compulsive disorders, and personality disorders. Several hypotheses have indeed been proposed to explain this relation. One of the first is by Caprara and Cervone [2], which states that the emotional information could become like an authoritative inner voice in people who chronically experience affective states such as anxiety, sadness, guilt, or fear [2]. While normal individuals would be able to eliminate or reduce the influence of the mood effect, clinical patients fail in stopping the inner voice from being experienced as authoritative (e.g., “If I feel anxious, then I blush” or “If I feel guilty, then I have to wash my hands 10 times before going outside”), which creates vicious circles, with the result of maintaining the beliefs and thus the disorder based on those beliefs.

More recently, Meeten and Davey [10] proposed the mood-as-input hypothesis (see above) as a theory that may be applied to further the understanding of perseveration across different disorders, including pathological worrying, chronic pain, compulsive checking, and depressive rumination. In particular, the mood-as-input hypothesis seems to predict that perseveration during a task may be influenced by the interactions between the individual’s stop rules and their concurrent mood. They also attest that the valence of an individual’s concurrent mood may be used as input or information about whether the stop-rule-defined goals for the task have been met. According to the authors, clinical individuals are characterized by specific features that tend to facilitate perseveration through mood-as-input processes.

Finally, more recently, Tiba [65] states that the emotion used as information can also be represented as mental pictures or images, which may influence the interaction of feelings and judgments, reinforcing the pathological beliefs from which the emotion arises and thus maintaining emotional disorders. Therefore, according to this approach, the emotional states can be co-opted as embodied simulations of feelings, that is, a partial re-creation of emotional states. This is to say that individuals with a personal aptitude for the creation of images or “mental movies” may be strongly influenced by the content of their imagination. According to this perspective, Berle and Moulds [56] would explain why depressed patients may not show increases in emotional reasoning as follows: these patients have difficulties in the simulation process (probably because of the lack of concentration).

In accordance with the literature (see above), we hypothesize that what contributes to the maintenance of psychological disorders is the individual relevance of trait-emotion in the use of state-emotion as information for inference or evaluations. For example, greater chronic anxiety or guilt (i.e., trait-emotion) leads to greater perceived risk or greater evaluation standards for performance. A relevant discrepancy between individuals scoring high and low in trait anxiety or guilt may consist of the faith in the informational value of guilty or anxious feelings. From this perspective, greater chronic anxiety or guilt may lead to greater perceived risk or standards. This would explain the reason why anxious, depressed, or obsessive patients tend to give great importance to the information produced by their temporary affect, which in turn would reinforce the beliefs (for example, of danger) at the basis of the emotional state itself, leading to the maintenance of the disorder.

The phenomenon similarly works in patients with personality disorders. However, in addition to the specific presence of a trait-emotion, patients with personality disorders are influenced by emotional reasoning for at least three other reasons: (1) the egosyntonic template, which prevents patients from looking at their responses in a critical way; (2) the higher sensitivity to emotional activity, which more deeply affects the cognitive control of negative emotional information (emotional events also usually trigger rumination and negative cognition; (3) the higher presence of information-processing biases, which distort most cognitive areas (including perception, attention, thought, memory, etc.).

In conclusion, considering the studies discussed in this paper, we suggest that emotional reasoning is one aspect of cognition that is reliably biased in many disorders. Moreover, negative emotional context, even context entirely unrelated to social or event judgments, may negatively influence the behavior and decision-making processes of patients. Therefore, we believe that an awareness of the cognitive biases triggered by emotional reasoning may result in new directions for treatment. Similarly, awareness of the impact of emotional context on these biases may help therapists and patients to intervene in automatic cognitive processes that perpetuate dysfunctional beliefs.

## 8. Limits of the Studies

In this paper we analyzed the phenomenon of emotional reasoning associated with major psychopathological disorders such as mood disorders, anxiety disorders, obsessive-compulsive disorders, and personality disorders. Although this study provides a well-structured review of the literature and a good description of the cognitive bias, thus contributing to a deeper understanding of this topic, the paper is a non-systematic review. It does not give an overview of the phenomenon in all psychopathological disorders, mainly because the topic has not yet been investigated on a whole series of psychopathologies, which are missing here. Furthermore, most of findings reported in our paper, which demonstrate the emotional reasoning mechanism in pathological disorders, have been obtained with analog samples. There are only a few trials that include individuals suffering from psychological disorders. This topic thus awaits further testing in clinical participants. However, the fact that non-clinical people with a general inclination to feel an emotion tend to use that state-emotion as information about, for example, the danger of a situation, even when the emotion has nothing to do with the situation, suggests that this may be a strong mechanism. The relation might be even stronger in a clinical group of subjects who experience that emotion chronically.

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
