# Peer review of "Emotional Reasoning and Psychopathology"

_brainsci, 2021, doi:10.3390/brainsci11040471_

Round 1

Reviewer 1 Report

I found this an interesting paper and enjoyed reading it. While I would not describe it as especially new thinking, it is a useful contribution to further thinking. Other researchers are cited extensively and their work explained. For me, it put into words something that I had known but not clearly expressed.

Summary of Content: The paper explores emotional reasoning and its effect on judgements about situations presented to individuals. It clarifies the role of emotional reasoning in the following disorders: anxiety, OCD and depression. Strengths: The paper is readable and sensibly organised. Weaknesses: None

Major improvements: None

Minor: There are a few uncertain words, for example, Line 103

Author Response

Point 1:  There are a few uncertain words, for example, Line 103

Response 1: Ex Line 103, now 98-99, was re-written in order to avoid uncertain words.

Point 2: Delete in the brackets of the references: "i.e.", "cf.", "see above"

Response 2: Abbreviations in the brackets of the references were deleted

Reviewer 2 Report

This non-systematic review analyzed the construct of emotional reasoning associated with major psychiatric disorders (i.e. mood disorders, anxiety disorders, OCD).

It is well-structured work and provides a good description of this cognitive bias.

I suggest some minor revisions:

- Delete in the brackets of the references: "i.e.", "cf.", "see above"

- In the section regarding OCD, please insert and discuss the results of the study: "If I feel disgusted, I must be getting ill": emotional reasoning in the context of contamination fear, Behav Res Ther. 2013 Mar; 51 (3): 122-7. doi: 10.1016 / j.brat.2012.11.005. Epub 2012 Dec 10.

- Please Insert a limits section at the end of discussion (i.e. non-systematic review)

- Page 6 line 284: please modify the typo error

Author Response

Point 1:  In the section regarding OCD, please insert and discuss the results of the study: "If I feel disgusted, I must be getting ill": emotional reasoning in the context of contamination fear, Behav Res Ther. 2013 Mar; 51 (3): 122-7. doi: 10.1016 / j.brat.2012.11.005. Epub 2012 Dec 10. -

Response 1: The article suggested was inserted and discussed in the section regarding OCD.

Point 2:  Please Insert a limits section at the end of discussion

Response 2: A limit section was added at the end of discussion

Point 3: Page 6 line 284: please modify the typo error

Response 3: the error was deleted

Reviewer 3 Report

This was an interesting review to read. Thank you very much.

I do not have much to add, but was wondering whether the review might additionally benefit from more explicitly incorporating an individual differences perspective? I think this could be possible in two ways.

First, I noticed that several clusters of disorders were discussed, but no personality disorders. However, emotional reasoning is also impaired there. It might thus broaden the paper to include personality psychopathology (in terms of categorical and dimensional models).

Second, in general individual differences in affect, affective processing, (different forms of) reasoning, and emotion regulation could be discussed a bit more. Surely, there are substantial and meaningful individual differences that will have consequences for treatment.

Author Response

Point 1: I do not have much to add, but was wondering whether the review might additionally benefit from more explicitly incorporating an individual differences perspective? I think this could be possible in two ways.

First, I noticed that several clusters of disorders were discussed, but no personality disorders. However, emotional reasoning is also impaired there. It might thus broaden the paper to include personality psychopathology (in terms of categorical and dimensional models).

Second, in general individual differences in affect, affective processing, (different forms of) reasoning, and emotion regulation could be discussed a bit more. Surely, there are substantial and meaningful individual differences that will have consequences for treatment.

Response 6: Individual differences were already discussed in the paper at lines 270-278/ 447-455/ 466-476. However, another new paragraph on personality disorders was added. In this paragraph were further explored the individual differences in individuals suffering for personality disorders.